# TARREAN: A Novel Transformer with a Gate Recurrent Unit for Stylized Music Generation

**DOI:** 10.3390/s25020386

**Published:** 2025-01-10

**Authors:** Yumei Zhang, Yulin Zhou, Xiaojiao Lv, Jinshan Li, Heng Lu, Yuping Su, Honghong Yang

**Affiliations:** 1School of Computer Science, Shaanxi Normal University, Xi’an 710062, China; zym0910@snnu.edu.cn (Y.Z.); zhouyulin@snnu.edu.cn (Y.Z.); lvxiaojiao@snnu.edu.cn (X.L.); l1062563242@163.com (J.L.); luheng698@163.com (H.L.); ypsu@snnu.edu.cn (Y.S.); 2Key Laboratory of Modern Teaching Technology, Ministry of Education, Shaanxi Normal University, Xi’an 710062, China; 3Key Laboratory of Intelligent Computing and Service Technology for Folk Song, Ministry of Culture and Tourism, Xi’an 710062, China

**Keywords:** automatic music generation, deep learning, transformer, gate recurrent unit, root mean square layer normalization

## Abstract

Music generation by AI algorithms like Transformer is currently a research hotspot. Existing methods often suffer from issues related to coherence and high computational costs. To address these problems, we propose a novel Transformer-based model that incorporates a gate recurrent unit with root mean square norm restriction (TARREAN). This model improves the temporal coherence of music by utilizing the gate recurrent unit (GRU), which enhances the model’s ability to capture the dependencies between sequential elements. Additionally, we apply masked multi-head attention to prevent the model from accessing future information during training, preserving the causal structure of music sequences. To reduce computational overhead, we introduce root mean square layer normalization (RMS Norm), which smooths gradients and simplifies the calculations, thereby improving training efficiency. The music sequences are encoded using a compound word method, converting them into discrete symbol-event combinations for input into the TARREAN model. The proposed method effectively mitigates discontinuity issues in generated music and enhances generation quality. We evaluated the model using the Essen Associative Code and Folk Song Database, which contains 20,000 folk melodies from Germany, Poland, and China. The results show that our model produces music that is more aligned with human preferences, as indicated by subjective evaluation scores. The TARREAN model achieved a satisfaction score of 4.34, significantly higher than the 3.79 score of the Transformer-XL + REMI model. Objective evaluation also demonstrated a 15% improvement in temporal coherence compared to traditional methods. Both objective and subjective experimental results demonstrate that TARREAN can significantly improve generation coherence and reduce computational costs.

## 1. Introduction

Music composition requires the creation of coherent musical segments, where the latter parts of a composition often refer to previous elements [1]. Repetition is a key aspect of music structure, ensuring coherence and consistency by systematically repeating themes and phrases. The use of repetition can create unique rhythms and variations in music. Computer-based music generation aims to simulate this systematic pattern of repetition [2]. Music generation involves two crucial factors: the method of encoding music into discrete symbol models and the algorithm used to model music sequences. To input music into the model, encoding in a format acceptable by the model is needed. Each note, rhythm, and chord in music data is mapped to a numerical representation, converting the digital representation of a musical score into a discrete symbol sequence arranged in chronological order, ultimately constructing music sequence data. Commonly used methods for encoding music data include MIDI-like, REMI, and compound word (CP) [3]. The key to music generation lies in using a network model to learn the dependencies between sequence elements. The model is trained to express the probability distribution of music sequences and sample from it to generate new musical compositions. By learning the structure and patterns of music, the model can generate coherent and creative musical segments. See Appendix A for details.

Mainstream music-generation methods can be categorized into two main types: statistical models and deep neural network models. Statistical models often employ the Markov chain method, where probabilistic models are created by analyzing large amounts of music data and statistical features of music clips are used to generate new music [4]. However, the generation of music through statistical models has inherent limitations. The Markov chain method is restricted to producing segments that are present in the original dataset, resulting in a lack of coherence in the generated music. In recent years, deep neural network models have gained popularity in music generation. These models are trained to learn the relationships between relevant musical features and notes, and then generate music based on the learned patterns. Examples of such models include RNN, LSTM, GAN, and other sequence models, which can simulate musical information and generate new music [5,6,7,8,9,10,11]. Transformer, a deep learning model based on self-attention mechanisms, has also been applied to music generation [12]. Its achievements in NLP tasks and a range of sequence-related tasks have introduced innovative concepts and approaches to music composition.

The self-attention mechanism has drawn more recognition in music generation because of its ability to model correlations at multiple timescales in long sequences. The attention mechanism has become an important component of long sequence modeling, which can model dependencies in various tasks without being limited by the distance between input and output sequences. In 2017, Google [10] presented a Transformer network model that entirely relies on attention mechanisms to capture global dependencies between inputs and outputs; it demonstrates great potential in time data modeling. Deng et al. [13] applied Transformer for the first time to music creation with a long-term structure, which shows significant application potential in music generation. The music Transformer [12] proposed by the Google Magenta team uses an autoregressive approach to generate new music segments by learning the probability distribution of musical sequences. The pop music Transformer [14] presented by Huang et al. uses a self-attention mechanism to learn long-range dependencies between notes and uses position encoding to process sequence information; popular piano music with a coherent structure for about a minute can be generated. Choi et al. [15] used the Transformer encoder and decoder to generate an accompaniment for a given melody. Wu et al. [16] used Jazz Transformer to generate jazz-style music. Donahue et al. [17] proposed the LakhNes model to generate multi-track music using Transformer.

The Transformer model is outstanding in long sequence memory tasks, and is widely used in music generation tasks [10]. Music creation involves not only the combination of notes and rhythm, but also factors such as emotion, expression, and personality, which are still difficult to be fully captured by the algorithm. Many studies have shown some of the limitations of the Transformer-based automatic music-generation method. First, since different types of music markers may have different properties, modeling that depends on these markers may be different to modeling based on words in the text. Secondly, problems, such as the music that is generated being too short and incoherent, may occur due to the limitations of the model. We can see in Table 1 a comparison of the existing methods.

To address the above problems, a TARREAN music melody generation method based on musical events is proposed in this paper; our main contributions include the following:(1)A TARREAN model for music generation is proposed, which replaces residual connection modules with GRU modules. It utilizes an update gate to control the iteration of old music sequence feature information and a reset gate to determine the degree of influence of old feature information on the training process in order to adapt to the current music characteristics. This improvement allows the TARREAN model to exhibit better performance in modeling long sequences and capturing dependencies in music.(2)Replacing layer norm with RMS norm can smooth gradients while reducing model training time. By propagating more gradient signals to early layers during network training, it helps the network to better learn the statistical characteristics of music sequences.

## 2. Data Pre-Processing and Data Representation

The TARREAN model mainly includes three parts: data pre-processing and representation, training and reasoning process of music generation network, and music evaluation. A framework of the method proposed is shown in Figure 1.

### 2.1. Musical Data Enhancement

To improve the generalization ability of the model, data-enhancement methods were used to increase the amount of data in the training set. We use random data enhancement; that is, each piece of data in the training set randomly selects a MIDI enhancement, as shown in Figure 2. Four kinds of MIDI data-enhancement methods are adopted in this paper.

(1)Pitch conversion: Create a new melody by converting the notes in the MIDI data through moving the pitch of each note up or down. For example, moving all notes up by a half tone or all notes down by a quarter to create a different yet still somewhat similar melody as to the original one, shown in Figure 2b.(2)Time offset: Make the length of the notes become longer or shorter by changing the MIDI duration value of the notes. Change the rhythm and performance of the music by adjusting the timestamps of the MIDI events. Random time offsets generate a random time offset for each MIDI event, which can control the range and distribution of random offsets. By introducing randomness, more variation and expressiveness can be added to the music, as shown in Figure 2c).(3)Add notes: Add additional notes to the existing MIDI data to increase the complexity and hierarchy of the music; this can extend the melody or harmony, as shown in Figure 2d.(4)Delete notes: Delete specific notes in MIDI data to reduce certain elements in music; this can make the music concise or change its structure, as shown in Figure 2e.

### 2.2. Data Representation of Music Event

There are three mainstream music representation methods: MIDI-like, REMI, and CP. The MIDI-like encodes information related to MIDI notes using “note on”, “note off”, and “velocity” markers, using the “time shift” marker to indicate the relative time interval between the two markers. REMI adopts a beat-based representation that uses the “bar” and “beat” markers to represent time information. The “bar” indicates the beginning of a new section, and “beat” points to one of the constant numbers of beat partitions in the section. In addition, REMI uses the “tempo” markers to control the rhythm of the music, and uses the “duration” marker to replace “note off”.

Both MIDI-like and REMI take the events as separate markers. In order to further reduce the length of the REMI-coded sequence, CP is proposed to convert REMI-coded discrete token sequences to CP representations. Here, the tokens belonging to the same family are grouped into a super-token and placed at the same time step. The CP considers three families by default: rhythm, note, and sequence end. Figure 3 illustrates the process of conversion from a REMI marker sequence to a compound word sequence.

As can be seen from Figure 3c, a compound word consists of grouping markers, ignoring markers, and a family marker, wherein color indicates grouping markers and white indicates ignoring markers. Family markers, N, are associated with notes, M represents markers related to measurement, and S indicates end of the sequence.

In the TARREAN music generation model, the CP sequence of music representation is encoded in the data input stage, and then converted to sparse vectors before input to the network.

## 3. Model Structure

Each input CP sequence is divided into fragments of compound word length, with a segment length of 512. Different feed-forward heads are used to model different types of events. The network utilizes the cross-entropy loss function to calculate the gradient loss, which is then clipped based on a maximum gradient norm. Finally, the Adam optimization algorithm is used to update the model’s parameters. After training is complete, the optimal model parameters from the training process are saved. During the music generation phase, the trained TARREAN method models the probability distribution of CP sequences. The next CP sequence is predicted using temperature sampling, finally generating new musical compositions.

### 3.1. The TARREAN Network Structure

In order to improve the effect of automatic music creation, TARREAN music generation network is proposed; architecture of the global model is shown in Figure 4.

The process of generating music using the TARREAN method involves both the training and generation phases of the model. Firstly, the music data undergo pre-processing, being converted into the CP encoding format. Each note, rhythm, and chord in the MIDI music data is mapped to a numerical representation, constructing the music sequence data. Throughout the training phase, the model undergoes iterative training and modifies its parameters through gradient descent and backpropagation to enhance the quality and diversity of the generated music sequences. In the generation phase, given a compound word as the input sequence, the model is invoked to generate the next compound word sequence.

The TARREAN music generation network model is based on the Transformer-XL architecture [18]. As an art form with a complex structure, music notes in the sequence consequently have a strong contextual correlation. A recursive mechanism is introduced to the music generation network model to process ultra-long music sequence data; its operation principle is shown in Figure 5. The introduction of the recursive mechanism in the network model allows for the utilization of the hidden state derived from the preceding segment to be stored as memory in the current segment, facilitating the establishment of feedback connections between segments. Such methods can learn dependencies over a fixed length and are able to capture ultra-long sequence dependencies, all the while maintaining temporal coherence.

Seven different music markers are encoded separately in the model; each musical event vector is represented by mapping to layers with an embedding matrix. The embedded sequences are then connected, and the connection results are linearized. The linearized results coupled with position encoding are fed to the network, and then dependencies among the sequences are captured by using the multi-head attention module. The proposed network model is composed of 12 layers of TARREAN modules shown in Figure 4, and each TARREAN module consists of multi-head attention unit with a mask, GRU, an RMS Norm unit, and a feed-forward neural network.

### 3.2. Multi-Head Attention Unit with Mask

Different musical genres are characterized by varying proportions of musical attributes. The application of the multi-head attention mechanism serves to optimize the processing of salient information; therefore, the effect of music generation can be optimized. In order to process the input sequences of musical events with an unfixed length $X=\left[x_{1},x_{2},\ldots ,x_{N}\right]$, the characteristics of the self-attention mechanisms are used to dynamically generate weights for different connections. The related vectors *Q*, *K*, and *V* are calculated based on Equations (1)–(3).(1)$$Q=W_{q}X$$(2)$$K=W_{k}X$$(3)$$V=W_{v}X$$
where $W_{q}{,W}_{k}$, and $W_{v\;}$ represent the weight matrix corresponding to the music. Equation (4) constitutes the computation involved in the self-attention mechanism.(4)$$Self\;Attention(Q,K,V)=soft\max\left(\frac{QK^{T}}{\sqrt{d_{k}}}\right)V$$
where $d_{k}$ represents the dimension of the key vector. The attention mechanism will input *X* in parallel to *N*, “head“ the attention mechanism calculation is conducted in each “head“ separately, the outputs of all the “heads“ are combined, and the final result of the multi-head attention mechanism is derived using an additional weight matrix $W_{o}$. Equations (5)–(7) illustrate the computation involved in the multi-head attention mechanism.(5)$$MultiHead\left(X\right)=\left[head_{1},head_{2},\ldots ,head_{N}\right]W_{o}$$(6)$$head_{n}=SelfAttention(Q_{n},K_{n},V_{n})$$(7)$$\forall n\in \left\{1,\cdots ,N\right\},Q_{n}=W_{q}^{n},K_{n}=W_{k}^{n},V_{n}=W_{v}^{n}$$
where $MultiHead\left(X\right)$ represents the final output of the multi-head self-attention mechanism, and ${head}_{n}$ is the output of each “head” after the self-attention mechanism.

In order to learn music characteristics from different dimensions and subspace, we use different parameters to conduct linear transformation to query vector *Q*, key vector, and value vector *V*, and input the results into the scalable point product attention to obtain different attention outputs; then, we substitute these outputs to Equations (5)–(7) to derive the final result. During the music reasoning process, the former part of the output should not refer to the element after that element; multi-head attention with masks is adopted to prevent the use of future-predicted information at the current moment.

### 3.3. GRU

Music is not just time sequence data: it also has a long contextual dependency. This means that changes in musical data are not only dependent on time, but are also affected by input data at a previous moment. In order to fully extract the time sequence and solve the long-distance dependence problem, the GRU module is introduced to the TARREAN model to control the music event information such as input and memory. The introduction of this module not only solves the gradient dispersion of the Transformer-XL network, but also simplifies the calculation of long-term and short-term memory unit, and the training efficiency of the music generation network is improved [19].

During the training process of the model, the update gate is used to control the iteration of the old music sequence, while the reset gate is used to control the influence of the old information in the calculation. The two-gate unit allows the GRU to better capture the long-term dependencies in the time sequence, so as to address the gradient disappearance problem of the Transformer-XL model in music generation [20].

With the proper use of the update and reset gate, GRU can ignore some irrelevant information without losing any information that is important. The retainment of useful information makes it more efficient than the traditional Transformer-XL model.

The internal calculation process in the GRU is shown in Equations (8)–(11).(8)$$z_{t}=\sigma (W_{z}\cdot [h_{t-1},x_{t}])$$(9)$$r_{t}=\sigma (W_{r}\cdot [h_{t-1},x_{t}])$$(10)$${\tilde{h}}_{t}=\tanh(W\cdot [r_{t}*h_{t-1},x_{t}])$$(11)$$h_{t}=(1-z_{t})*h_{t-1}+z_{t}*{\tilde{h}}_{t}$$

Equation (8) is the update gate; it uses the sigmoid function σ to control the impact of the hidden state $h_{t-1}$ at the previous moment and the music sequence input information $x_{t}$ at the current moment on the hidden state $h_{t}$ at the next moment. The update gate ranges inside 0–1, and can decide which information is retained or discarded. Equation (9) is the reset gate; the sigmoid function σ is used to control the impact of the hidden state $h_{t-1}$ at the previous moment and the music sequence $x_{t}$ at the current moment on the candidate hidden state ${\tilde{h}}_{t}$. The reset gate can decide whether to ignore past information and better accommodate the musical characteristics at the current moment. Equation (10) is the calculation of the candidate hidden state; it applies the linear transformation weight matrices$\;W_{z}$, $W_{r}$, and W to the input information at the current moment music sequence $x_{t}$, and the reset gate $r_{t}$ to the hidden state $h_{t-1}$ at the previous moment. This computational approach can generate a candidate hidden state ${\tilde{h}}_{t}$ that contains information about past hidden states and current music input. Equation (11) is the expression of update memory; it uses update gate $z_{t}$ to determine how many hidden states $h_{t-1}$ and candidate hidden states ${\tilde{h}}_{t}$ at the previous moment are retained, and combines them with the hidden state $h_{t}$ at the current moment. This operation can dynamically update and memorize the music sequences in order to better generate coherent music.

### 3.4. RMS Norm Unit

The input distribution of each layer may change with the training process and this unstable input distribution calls for the RMS Norm unit [21] that can normalize the characteristics of each music data sample and reduce the distribution variation, which can improve stability and training speed [22]. RMS Norm can enable more gradient signals to be propagated to the early network layer, help the network to better learn the statistical characteristics of the input data, and improve the convergence and training efficiency of the music generation network.

RMS Norm uses the root mean square to normalize, which can reduce the impact of noise. The calculation Equation is shown in Equation (12).(12)$${\bar{a}}_{i}=\frac{a_{i}}{RMS(a)}g_{i},\;where\;RMS(a)=\sqrt{\frac{1}{n}\sum_{i=1}^{n}a_{i}^{2}}$$
where $a_{i}$ is the musical sequence data input, RMS represents the calculation function of root mean square, and $g_{i}$ is the gain parameter for the re-scaled standardized sum input, which is set to 1 at the beginning.

In the music generation task, using RMS Norm can help the model to converge better and faster, making the generated music more natural. It makes model training more stable, shortens the training time, and improves the quality of the generated music.

The TARREAN model is mainly composed of the above three modules. We obtain the output sequence of the multi-head attention module first, and then the embedded input sequence passing the position code is fused with the output sequence of multi-head attention module to avoid the gradient dispersion by using GRU connection. Then, RMS Norm standardization is conducted to the fused results, and the specific calculation Equation is shown in Equation (13).(13)y=RMS(gate(x,Attention(Q,K,V)))
where y represents the output after standardization function calculation, the RMS represents RMS Norm function, gate represents the gate controlled GRU connection function, x represents the music sequence input, and $Attention\left(Q,K,V\right)$ represents the computational function of attention. Q,K,V are the query vector, key vector, and value vector in the attention mechanism.

The fused results are fed to the feed-forward network layer for further processing. This encoding process is repeated multiple times, and then the output of the last encoding layer is subjected to a linear regression operation. Finally, after processing by the Softmax layer, the probability distribution of the notes can be obtained as the result of the output layer. Then, the output sequence is obtained, and it can be converted into the corresponding musical sequence.

## 4. Experiments and Results Analysis

### 4.1. Dataset and Experimental Setup

The ESAC (Essen Association Code and Folk Song Database) folk song dataset [23] collected 20,000 melodies, most of which are from Germany, Poland, and China. We selected 802 Shanxi folk songs from the ESAC dataset as our experimental dataset.

The model was implemented in the Python 3.6 environment, using the PyTorch framework, on the GeForce RTX 2080TI platform. In the process of training the deep learning model, the cross-entropy function is used as a loss function to measure the difference between the predicted probability distribution and the true probability distribution, and the smaller cross-entropy value indicates a better prediction effect [24]. In order to reduce the loss value gradually, Adam optimizer [25] is used; the initial learning rate is 0.00001, the batch size is set to 4, and the training is set to end after 1200th rounds, with the weight and bias terms in the network model adjusted through back-propagation. After the operation outlined above, the training of the TARREAN music melody automatic generation model was finally completed.

### 4.2. Model Performance Comparison

To evaluate the effect of RMS Norm, we compare the model training time using RMS Norm and layer norm. The comparison results of model training time using RMS Norm and layer norm are shown in Table 2.

TARREAN’s integration of the GRU modules addresses the limitations of traditional Transformer-based methods by providing better temporal modeling for ultra-long sequences. The GRU’s update and reset gates ensure that essential features from prior sequences are retained while irrelevant information is discarded. This mechanism resolves issues of gradient vanishing and incoherent sequence generation often encountered in Transformer-based models.

RMS Norm normalizes the input features using a root mean square operation rather than the mean and variance calculations in layer norm. This operation has a complexity of O(n). In comparison, layer norm requires calculating both the mean and variance, each with a complexity of O(n), resulting in an overall complexity of O(2n).

As can be seen from Table 1, the training time of the Transformer-XL + RMS + CP model is 0.88% less than that of Transformer-XL + CP, and the training time of TARREAN is 22.80% less than that of Transformer-XL + GRU + CP, which indicates that, compared to the layer norm, the RMS Norm can reduce the model training time.

### 4.3. Objective Evaluation Experiments and Results

To verify the music generation effect of the TARREAN music auto-generation network model, objective and subjective evaluation experiments are conducted to evaluate the similarity of the music generated by using five models: Transformer-XL + REMI, Transformer-XL + CP, Transformer-XL + RMS + CP, Transformer-XL + GRU + CP, and TARREAN.

#### 4.3.1. Objective Evaluation Index

The objective evaluation of music generation serves as a valuable method for assessing the music generation task. A framework for objective evaluation has been developed to assess the effectiveness of the TARREAN model by examining three dimensions: pitch, rhythm, and musical structure.

**(1)** 
**Index related to pitch**


Pitch range (PR): indicates the range of pitch.

Pitch-scale ratio (PSR): indicates the ratio of pitch change within a scale.

Pitch entropy (PE): a measure of diversity or uncertainty of pitch in music. It measures the distribution of pitch in music, i.e., the degree of change in pitch.

Scale consistency (SC): determined by calculating the ratio of pitches across all standard scales and reporting the proportion of the most optimal matching scale.

Pitch histogram entropy (PH): a concept from information theory that quantifies the probability distribution [26] and serves as an indicator for measuring pitch distribution in music. By gathering the notes within a given section, a 12-dimensional pitch histogram $\vec{h}$ is constructed based on the pitches (e.g., C, C#, A#, B). This method allows for the exploration of various pitches by normalizing the total number of notes within the sections. The entropy of the pitch histogram $\vec{h}$ can be computed using Equation (14).(14)$$H(\vec{h})=-\sum_{i=0}^{11}h_{i\log_{2}(h_{i})}$$

**(2)** 
**Index related to rhythm**


Empty beat rate (EBR): refers to the ratio of empty beats within the rhythmic structure of the music.

Grooving Pattern similarity (GS): a rhythm pattern similarity index utilized to assess the rhythmic elements present in the music. The rhythmic pattern represents the position in the section where at least one of the notes starts; it is represented as $\vec{g}$ Equation (15) defines the similarity of a pair of rhythmic patterns ${\vec{g}}^{a}$ and ${\vec{g}}^{b}$, wherein *Q* represents the dimensions of ${\vec{g}}^{a}$ and ${\vec{g}}^{b}$, $XOR\left(.,.\right)\;$represents the sum and different operation, and the values of $GS\left(.,.\right)\;$are always between 0 and 1.(15)$$GS({\vec{g}}^{a},{\vec{g}}^{b})=1-\frac{1}{Q}\sum_{i=0}^{Q-1}XOR(g_{i}^{a},g_{i}^{b})$$

**(3)** 
**Structural indicators**


The structure of music is formed by the repetitive musical content in the work; it involves multiple levels/phases, from the immediate musical idea to the entire movement [27]. From a psychological perspective, the structure of the repetition creates the fascinating and evocative nature of music.

The structural indicators presented in this study draw on an adapted figure [28] and are designed to identify the most prominent repetitive sections within a specified time interval, thereby reflecting the repetition structure in music. To simplify the mathematical relation, assuming a sampling frequency of 1 Hz for the adaptive graph matrix *S*, the structural indicators are defined in Equation (16).(16)$$SI_{l}^{u}(S)=\underset{\begin{array}{l} l<i<u\\ 1\leq j\leq N \end{array}}{\max}S_{i,j}$$
where in l,u N are the up and down limits of the fragment time interval (in seconds), and N is the duration of the segment (in seconds). In the experiments, we select $SI_{3}^{8}$, $SI_{8}^{15}$, and $SI_{15}$, which denote the structural indicators that represent the short-term, medium-term, and long-term structures of music, respectively.

#### 4.3.2. Experimental Results and Analysis

The trained model was used for music generation. Each model is used to generate 50 songs; these songs are extracted into the training set and the mean index values of 50 songs are found for each model. Table 3 presents the mean values of the music generated by each model across the various evaluation dimensions.

To visually assess the similarity between generated music and the training set, the differences in each evaluation metric were calculated for music produced by each model compared to the training set, as shown in Table 3.

In Table 3 and Table 4, the music generated by the five different models was compared with the music in the training set by 10 evaluation indicators (PR, PSR, PE, PH, SC, EBR, GS, SI_short, SI_mid, and SI_long). The results show that the music obtained using the TARREAN model has the smallest difference from the training set in seven indexes: PR, PSR, PE, GS, SI_short, SI_mid, and SI_long, which is better than the other four models. Therefore, the musical structure generated by the model in this paper is closer to that in the training set.

### 4.4. Subjective Evaluation Experiment and Results

A hearing test was conducted for subjective evaluation, where participants rated the music based on their subjective impressions. Three songs were randomly selected from the training set, along with music generated by five different models, making a total of eighteen pieces of music for the experiment. Participants were asked to evaluate each piece on a 1 to 5 scale, with higher scores indicating a more favorable impression. The evaluation was based on various subjective factors such as melody, harmony, rhythm, emotional expression, and overall enjoyment. A total of 30 subjects participated in the experiment, including 10 music professionals and 20 non-professionals. The outcomes of this subjective assessment are shown in Figure 6 and Table 5.

As shown in Figure 6, the TARREAN model achieved the highest satisfaction score for generated music, close to the score of the music in the training set. Subjective satisfaction scores in Table 5 reveal that the average rating for music produced by the Transformer-XL + REMI model was only 3.79, while the TARREAN model received an average rating of 4.34. This suggests that the music generated by the TARREAN model aligns more closely with listeners’ preferences.

## 5. Conclusions

In order to enhance the effectiveness of music generation, this paper proposes a TARREAN music melody generation method based on music events to address the issues of a lack of coherence and fluency, as well as low computational efficiency in music generation. This method utilizes Transformer-XL as the backbone network and introduces a GRU module to enable the model to capture dependencies in long music sequences. Additionally, the RMS Norm technique is employed to smooth gradients, simplify computations, and reduce model training time. The generated music is evaluated through subjective and objective evaluation experiments. The experimental results demonstrate that the proposed method achieves better music generation performance and significantly improves the efficiency of model training. Future research could further explore other improvement techniques and music data representations to further enhance the quality and diversity of music generation.

Although this research has achieved significant advancements in music generation, there are still several avenues for future exploration. One promising direction is the integration of sensor technologies with the TARREAN model to enhance the interactivity and adaptability of generated music. By incorporating real-time data from sensors such as biosensors (e.g., heart rate monitors) or motion sensors, the model can dynamically adjust the rhythm, tempo, and emotional expression of the music based on the user’s physiological or environmental state. For example, biosensor data can be used to influence the emotional tone of the music, while motion sensor data can alter the tempo to synchronize with the user’s movements, creating a more personalized and immersive experience.

This sensor-based interactive music generation could have a wide range of practical applications, including personalized music creation in fitness and wellness apps, as well as interactive art installations in smart homes or museums. Future research could explore the efficient integration of various sensor data types with the TARREAN model, which would further enhance the quality and diversity of generated music while providing a more engaging and adaptive user experience.

## Figures and Tables

**Figure 1 sensors-25-00386-f001:**
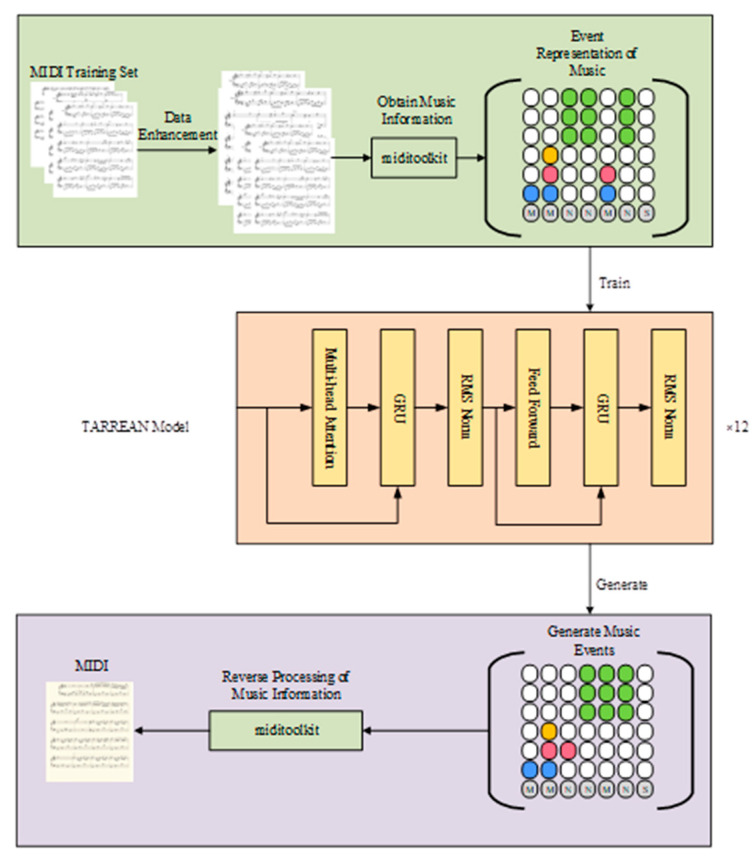
Framework of the TARREAN model based on musical events.

**Figure 2 sensors-25-00386-f002:**
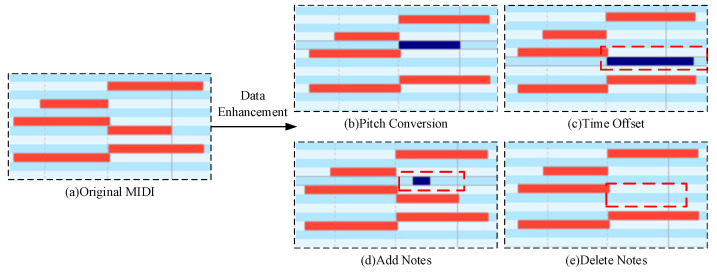
MIDI file fragments after data enhancement. Red bars represent the original MIDI notes, while blue bars indicate notes that have been modified during the data enhancement process, such as through pitch conversion, time offset adjustment, note addition, or note deletion. Dashed red boxes highlight specific regions where notes have been added or deleted.

**Figure 3 sensors-25-00386-f003:**
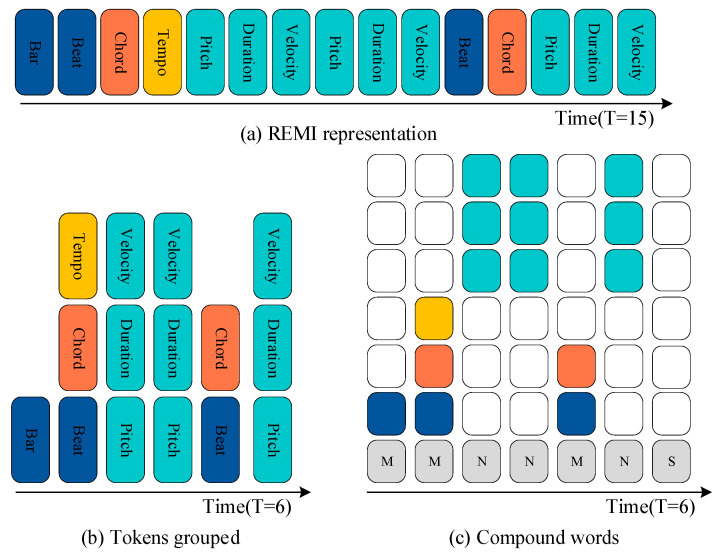
Transformation and representation of compound words sequences.

**Figure 4 sensors-25-00386-f004:**
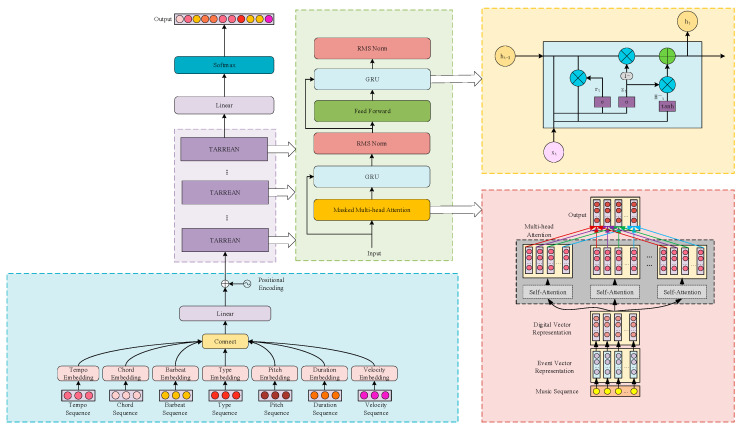
Architecture of TARREAN.

**Figure 5 sensors-25-00386-f005:**
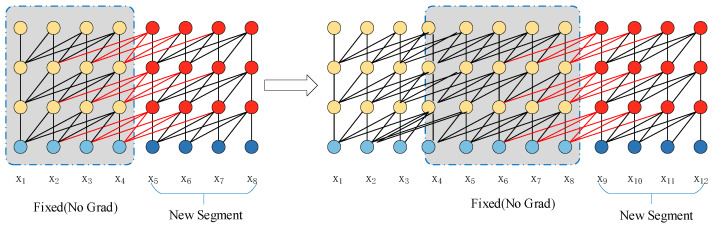
Recursive mechanism in Transformer-XL.

**Figure 6 sensors-25-00386-f006:**
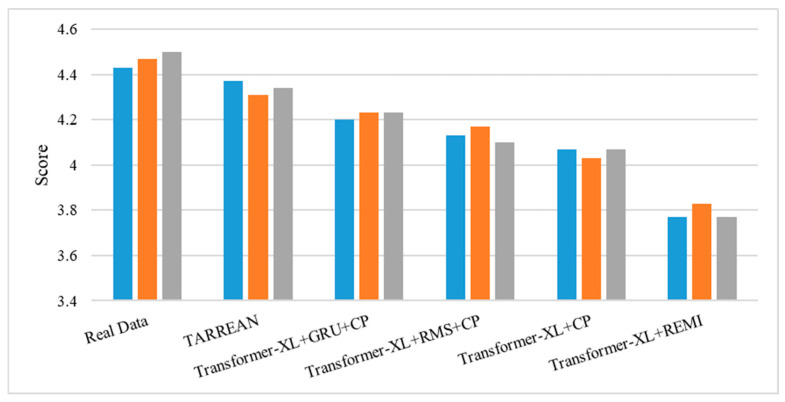
Satisfaction rating results of music generated by each model.

**Table 1 sensors-25-00386-t001:** A detailed comparison of the features and limitations of existing methods with TARREAN.

Method	Strengths	Limitations
Music Transformer	Long-range dependency modeling	Lacks temporal coherence in ultra-long sequences; high computational cost
Pop Music Transformer	Beat-based modeling; expressive output	Limited scalability; requires extensive data
Transformer + GRU Hybrid	Improved coherence; contextual learning	Limited applications in music generation

**Table 2 sensors-25-00386-t002:** Training run times for the different models.

Model	Training Duration (Days)
Transformer-XL + CP	2.27
Transformer-XL + RMS + CP	2.25
Transformer-XL + GRU + CP	3.07
TARREAN	2.37

**Table 3 sensors-25-00386-t003:** Mean values of evaluation metrics for each dimension.

Model	Pitch-Related					Rhythm-Related		SI		
	PR	PSR	PE	PH	SC	EBR	GS	SI_short	SI_mid	SI_long
Real Data	16.31	0.8724	2.7804	2.3313	0.9991	0.0574	0.9353	0.7077	0.6891	0.6705
Transformer-XL + REMI	30.10	0.8857	4.0688	3.1572	0.8773	0.2568	0.8674	0.6405	0.6086	0.6027
Transformer-XL + CP	29.42	0.8871	4.0254	3.1427	0.8877	0.2187	0.8678	0.6399	0.6161	0.6044
Transformer-XL + RMS + CP	30.94	0.8810	3.9987	3.1453	0.8816	0.2525	0.8674	0.6360	0.6118	0.6108
Transformer-XL + GRU + CP	31.06	0.8934	4.0038	3.1234	0.8936	0.2542	0.8651	0.6344	0.6158	0.6130
TARREAN	29.10	0.8674	3.9357	3.1764	0.8694	0.2580	0.8714	0.6462	0.6289	0.6169

**Table 4 sensors-25-00386-t004:** Absolute differences between generated music data and training set data.

Model	Pitch-Related					Rhythm-Related		SI		
	PR	PSR	PE	PH	SC	EBR	GS	SI_short	SI_mid	SI_long
Transformer-XL + REMI	13.79	0.0133	1.2884	0.8259	0.1218	0.1994	0.0679	0.0672	0.0805	0.0678
Transformer-XL + CP	13.11	0.0147	1.2450	0.8114	0.1114	0.1613	0.0675	0.0678	0.0730	0.0661
Transformer-XL + RMS + CP	14.63	0.0086	1.2183	0.8140	0.1175	0.1951	0.0679	0.0717	0.0773	0.0597
Transformer-XL + GRU + CP	14.75	0.0210	1.2234	0.7921	0.1055	0.1968	0.0702	0.0733	0.0733	0.0575
TARREAN	12.79	0.0050	1.1553	0.8451	0.1297	0.2006	0.0639	0.0615	0.0602	0.0536

**Table 5 sensors-25-00386-t005:** Subjective evaluation of experimental results.

Average Scores	Real Data	Transformer-XL + REMI	Transformer-XL + CP	Transformer-XL + RMS + CP	Transformer-XL + GRU + CP	TARREAN
Professionals	4.43	3.60	4.03	4.07	4.20	4.28
Non-professionals	4.48	3.88	4.07	4.17	4.23	4.37
All participants	4.47	3.79	4.06	4.13	4.22	4.34

## Data Availability

The dataset used in this study is derived from the ESAC (Essen Associative Code and Folk Song Database) folk song collection, which includes 20,000 melodies from Germany, Poland, and China. We selected 802 Shanxi folk songs from the ESAC dataset as our experimental data. Access to the original dataset is subject to third-party licensing and copyright restrictions. The ESAC dataset is maintained by the Institute of Arts, Polish Academy of Sciences. For access to the dataset and inquiries about usage permissions, please contact Ewa Dahlig, Deputy Director of the Institute of Arts, Polish Academy of Sciences, at the following email address: dahlig-turek@ispan.pl.

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
