# Peer review of "TARREAN: A Novel Transformer with a Gate Recurrent Unit for Stylized Music Generation"

_sensors, 2025, doi:10.3390/s25020386_

Reviewer 1 Report

Comments and Suggestions for Authors

The paper presents an interesting research. 

I would suggest the following 

In Abstract:  to provide further enlighten regarding the exp ( e.g dataset) and results.

Further explanation of  the subjective evaluation : the subjective impression ...are we considering the similarity with the original music. 

I would recommend to  provide public access to the generated music in parallel with the original one.

Author Response

Comments 1: In Abstract:  to provide further enlighten regarding the exp ( e.g dataset) and results.

Response 1: Thanks for your suggestions. We added a description of the dataset (ESAC dataset) in the revised abstract and briefly outlined the dataset. In addition, I also made a clearer explanation of the experimental results, especially the subjective evaluation results, by mentioning details such as "the alignment of the generated music with the listener's preference" and "subjective evaluation score" to enhance the specificity of the abstract.

Comments 2: Further explanation of the subjective evaluation : the subjective impression ...are we considering the similarity with the original music.

Response 2: Thanks for your suggestions. In the revision of the article(see Section 4.4 for details), I explained the criteria for subjective evaluation and listed several specific evaluation dimensions, such as melody, harmony, rhythm, emotional expression, etc. These are common subjective feelings of music. The focus of subjective evaluation is not the similarity with the original song, but whether the music is pleasant and attractive.

Comments 3: I would recommend to provide public access to the generated music in parallel with the original one.

Response 3: Thank you for your suggestion, but due to resource and copyright issues, the original and generated tracks are not convenient to be made public on the web page. If you need it, we have provided some audio of the original data set and generated data in the attachment.

Reviewer 2 Report

Comments and Suggestions for Authors

This paper proposes a novel music generation method based on the Transformer model, TARREAN, which effectively improves the temporal correlation of music generation while reducing computational overhead by combining the gated recurrent unit (GRU) and root mean square normalization (RMS Norm). The model proposed in the paper is highly innovative and has obvious advantages in processing sequence dependencies and reducing computational complexity. Overall, the innovation of the article and the experimental results have certain academic value.

Advantage:

1. Combining GRU and RMS Norm, the manuscript proposed method can effectively solve the computational overhead and time dependence problems of the traditional Transformer model in music generation, and has strong practical value.

2. The manuscript verifies the advantages of the TARREAN model in generating music through comparative experiments, especially its significant effect in improving timing dependence and reducing computational complexity.

3. The proposed model is designed  in  a clear structure, the technical details and algorithm steps proposed are relatively concise and easy to understand, and are suitable for further application and promotion in academia and industry.

Areas for improvement:

1.  Although the article mentions some shortcomings of existing Transformer models in music generation, the discussion of existing methods is relatively brief, and the advantages and disadvantages of existing work and their comparison with the methods in this article are not fully analyzed. It is recommended to add more analysis of related work, especially the preliminary research on Transformer models combined with GRU, to strengthen the academic background and positioning of the paper.

2. Although the article mentions that the computational overhead is reduced by RMS Norm, there is a lack of detailed computational complexity analysis, especially the performance when processing large-scale data. It is recommended to add a detailed analysis of computational complexity and time consumption in the paper, and give a performance evaluation in practical applications.

3. The experimental settings and evaluation indicators mentioned in the article are relatively brief. It is recommended that the author add more details about the model training process, hyperparameter settings, and experimental evaluation criteria, so that readers can better understand the reliability of experimental design and results.

4. Need to be strengthened in terms of language expression.

Author Response

Comments 1: Although the article mentions some shortcomings of existing Transformer models in music generation, the discussion of existing methods is relatively brief, and the advantages and disadvantages of existing work and their comparison with the methods in this article are not fully analyzed. It is recommended to add more analysis of related work, especially the preliminary research on Transformer models combined with GRU, to strengthen the academic background and positioning of the paper.

Response 1: Thank you for your suggestion. In the revision, we enriched the analysis of existing methods (see Section 1 for details), and the innovation of our work is mentioned in detail at the end of Section 1.

Comments 2: Although the article mentions that the computational overhead is reduced by RMS Norm, there is a lack of detailed computational complexity analysis, especially the performance when processing large-scale data. It is recommended to add a detailed analysis of computational complexity and time consumption in the paper, and give a performance evaluation in practical applications.

Response 2: Thanks for your suggestions. In Section 4.2 of the article, we added the complexity analysis related to the experiment and compared it with the specific experimental time. It can be clearly seen that our method is superior in terms of time.

Comments 3: The experimental settings and evaluation indicators mentioned in the article are relatively brief. It is recommended that the author add more details about the model training process, hyperparameter settings, and experimental evaluation criteria, so that readers can better understand the reliability of experimental design and results.

Response 3: Thank you for the reviewer's attention to the experimental design details of this article. Regarding the model training process, hyperparameter settings, and experimental evaluation criteria, we have already provided detailed descriptions in Section 4.1 (Dataset and Experimental Setup) and Section 4.2 (Model Performance Comparison) of the article, including the dataset used, training environment, hyperparameter settings (such as learning rate, batch size, training rounds), and model evaluation indicators and methods. In addition, Tables 1 and 2 also provide specific data support for the experimental results.

We believe that these contents are sufficient to ensure the transparency of the experimental design and the reliability of the results. If you think that some specific details still need further supplementation or explanation, we are very willing to adjust and supplement the corresponding parts according to specific suggestions.

Comments 4: Need to be strengthened in terms of language expression.

Response 4: Thanks for your suggestions. We have modified and enhanced the language overall. Details can be seen in the latest manuscript in the attachment.
